# *Streptomyces rochei* MS-37 as a Novel Marine Actinobacterium for Green Biosynthesis of Silver Nanoparticles and Their Biomedical Applications

**DOI:** 10.3390/molecules27217296

**Published:** 2022-10-27

**Authors:** Sobhy E. Elsilk, Maha A. Khalil, Tamer A. Aboshady, Fatin A. Alsalmi, Sameh S. Ali

**Affiliations:** 1Botany and Microbiology Department, Faculty of Science, Tanta University, Tanta 31527, Egypt; 2Biology Department, College of Science, Taif University, P.O. Box 11099, Taif 21944, Saudi Arabia; 3Oral and Maxillofacial Surgery and Diagnostic Sciences, Faculty of Dentistry, Taif University, Taif 21944, Saudi Arabia; 4Biofuels Institute, School of the Environment and Safety Engineering, Jiangsu University, Zhenjiang 212013, China

**Keywords:** marine actinobacteria, silver nanoparticles, antibacterial, anti-inflammatory, antibiofilm, cytotoxicity

## Abstract

Periodontitis, as one of the most common diseases on a global scale, is a public health concern. Microbial resistance to currently available antimicrobial agents is becoming a growing issue in periodontal treatment. As a result, it is critical to develop effective and environmentally friendly biomedical approaches to overcome such challenges. The investigation of *Streptomyces rochei* MS-37’s performance may be the first of its kind as a novel marine actinobacterium for the green biosynthesis of silver nanoparticles (SNPs) and potentials as antibacterial, anti-inflammatory, antibiofilm, and antioxidant candidates suppressing membrane-associated dental infections. *Streptomyces rochei* MS-37, a new marine actinobacterial strain, was used in this study for the biosynthesis of silver nanoparticles for various biomedical applications. Surface plasmon resonance spectroscopy showed a peak at 429 nm for the SNPs. The SNPs were spherical, tiny (average 23.2 nm by TEM, 59.4 nm by DLS), very stable (−26 mV), and contained capping agents. The minimum inhibitory concentrations of the SNPs that showed potential antibacterial action ranged from 8 to 128 µg/mL. Periodontal pathogens were used to perform qualitative evaluations of microbial adhesion and bacterial penetration through guided tissue regeneration membranes. The findings suggested that the presence of the SNPs could aid in the suppression of membrane-associated infection. Furthermore, when the anti-inflammatory action of the SNPs was tested using nitric oxide radical scavenging capacity and protein denaturation inhibition, it was discovered that the SNPs were extremely efficient at scavenging nitric oxide free radicals and had a strong anti-denaturation impact. The SNPs were found to be more cytotoxic to CAL27 than to human peripheral blood mononuclear cells (PBMCs), with IC_50_ values of 81.16 µg/mL in PBMCs and 34.03 µg/mL in CAL27. This study’s findings open a new avenue for using marine actinobacteria for silver nanoparticle biosynthesis, which holds great promise for a variety of biomedical applications, in particular periodontal treatment.

## 1. Introduction

Periodontitis is an inflammatory disease that affects the teeth’s supporting structures, resulting in the gradual deterioration of periodontal tissues, loss of attachment, aesthetics, and, eventually, tooth loss [1]. Even though there are numerous causes of periodontitis, bacterial plaque is frequently identified as the primary etiological agent of this oral disease [2]. Dental plaque is a biofilm formed by bacteria on the surface of teeth, gingiva, and restorative or prosthetic materials [3]. *Staphylococcus* spp., *Streptococcus mutans*, *Porphyromonas gingivalis*, and *Aggregatibacter actinomycete mcomitans* are among the bacteria implicated in the etiology of periodontitis, with *Staphylococcus* being the most common bacterium responsible for microbial infections associated with biofilms [4]. Periodontal treatment’s major objective is to destroy the microbial biofilm and decrease inflammation to establish and maintain sufficient infection control [5]. Numerous antimicrobial medicines have been utilized to treat periodontal disease. Mechanical removal of plaque and frequent application of systemic and topical antibacterial medications are only partially successful against the microorganisms responsible for periodontal diseases [6]. As a result, various agents with advanced physicochemical characteristics should be investigated, emphasizing antibacterial agents with innovative and distinct features that may be utilized as a substitute for periodontal treatments. Periodontal regeneration requires the isolation of gingival epithelial and connective tissue cells from the injured area, which results in the invention and implementation of guided tissue regeneration (GTR) membranes [7]. The regeneration of various intrabody lesions has been accomplished with effectiveness and dependability using GTR procedures using non-absorbable and bioabsorbable membranes. Rossa et al. [8] emphasized the importance of containing or eliminating periodontal infections for barrier membranes to reattach. Several techniques for managing or eradicating periodontal infections during GTR treatments have been promoted [9,10].

Inflammation is a physiological response to potentially harmful stimuli such as irritants, damaged tissue, or infections [11]. Systemic or localized inflammation can be acute or chronic [12]. Numerous mediators, such as prostaglandins, cytokines, and various reactive oxygen species (ROSs), such as nitric oxide (NO), are produced by various immune cell types or neutrophil respiratory bursts to protect cells and tissues during the acute inflammatory phase [13]. Synthetic medicines, formerly extensively used to treat inflammation, are no longer safe due to drug-related toxicity, iatrogenic responses, and dangerous adverse reactions that hinder therapy progression when administered long-term [14]. A safer and more effective alternative to conventional medicine, which has demonstrated efficacy in treating a variety of human disorders over the past several decades, must be developed as a result.

Recently, nanomaterials as treatments have emerged as a novel strategy for preventing and controlling the spread of many serious diseases [15,16,17,18,19]. Silver nanoparticles (SNPs) have displayed remarkable biocidal properties against a variety of pathogens, including some oral bacteria [20,21,22,23], and have even demonstrated superior antimicrobial properties to dental antiseptic solutions, which are considered essential in a dental clinic [24]. SNPs possess unique optical, electromagnetic, catalytic, and electrical characteristics, resulting in their extensive application as antimicrobial, anti-inflammatory, and anticancer medicines [25,26,27]. SNPs have been synthesized using a variety of methods, including classical (physical and chemical) and biological procedures [28,29,30]. Researchers have employed extreme reaction tracking to determine the difference between the biochemical reduction of nanoparticles using green production and a conventional approach [31,32,33,34]. Green nanoparticles had considerably lower cytotoxicity than chemical nanoparticles, suggesting that they are safe and may be used widely in biomedical applications [17,18,35]. For the manufacture of nanoparticles, green synthesis techniques such as bacteria, in particular actinobacteria, fungi, yeast, and plants, may be employed [36,37]. Amongst them, actinobacteria are a frequent source of SNPs with anticancer, antioxidant, and antimicrobial properties [21].

Several studies have recently focused on silver nanoparticles, and few studies have been published on actinobacteria, in particular *Streptomyces rochei*, with the ability to produce nanoparticles [38]. However, the effectiveness of SNPs in periodontal therapy, notably marine actinobacteria, remains unexplored or at its early stage. To the authors’ knowledge, no studies have been undertaken so far to investigate the performance of *Streptomyces rochei* MS-37 as a new marine actinobacterium for green biosynthesis of SNPs valued for antibacterial action against gingival pathogens and their effectiveness in periodontal treatment. Therefore, the current research focuses on the production of silver nanoparticles using *Streptomyces rochei* MS-37 and the elucidation of their antibacterial efficacy against oral pathogenic bacterial strains. In addition, the effectiveness of bio-SNPs’ antibiofilm and anti-inflammatory potential and antioxidant possibilities in decreasing membrane-associated dental infections was evaluated.

## 2. Materials and Methods

### 2.1. Cell Culture

Cell culture flasks (75 cm^3^, 250 mL volume) with 10% fetal bovine serum (FBS; Life Technologies, Gibco^®^; Carlsbad, CA, USA) and 50 g/mL gentamicin were used to cultivate the CAL27 oral adenosquamous carcinoma cells. Cells were cultured in Dulbecco’s modified Eagle’s medium (DMEM; Life Technologies, Gibco^®^; Carlsbad, CA, USA) (Naviform, Annapolis, GO, Brazil). Daily, cultures were examined under an inverted microscope while being incubated in 37 °C incubators with 5% CO_2_. When cell growth achieved a confluence of 70–80% of the total volume of the culture flask, cells were detached using trypsin. Centrifugation was used to separate human peripheral blood mononuclear cells (PBMCs) using a Ficoll density gradient (Ficoll-Paque Plus; GE Healthcare Bio-Sciences AB; Chicago, IL, USA). After two washes with saline, cells (0.3 × 10^6^ cells/mL) were resuspended in RPMI media supplemented with 20% FBS, 2 mM glutamine, and 50 μg/mL gentamicin.

### 2.2. Nanoparticle Biosynthesis

Actinobacterial strain MS-37 was cultivated for 7 days at 28 °C on a starch casein agar medium [39]. MS-37 was molecularly characterized using 16S rDNA gene sequencing, as published by Khalil et al. [22]. The actinobacterial biomass was centrifuged at 6000× *g* for 10 min, cleaned three times in sterile distilled water, and then autolyzed for three days at 28 °C in the same sterile distilled water. The cell debris was removed by centrifugation at 6000 rpm for 15 min, and the secondary metabolites of the actinobacterium were combined with silver nitrate (AgNO_3_; 1 mM final concentration; Sigma Aldrich) as a precursor to create SNPs. A 20 μL aliquot of overnight-grown *Streptomyces rochei* MS-37 was inoculated into 100 mL of nutrient broth medium, which was subsequently incubated on an orbital shaker at 28 °C and 120 rpm for 96 h [40]. After incubation, the actinobacterium biomass was collected by 30 min of centrifugation at 5000 rpm. To remove associated media components, 2 g of cells was washed twice with distilled water. The biomass was reconstituted in 150 mL of deionized water and stored at 28 °C for 48 h. Bacteria were lysed osmotically, then filtered via Whatman No. 1 filter paper. As previously reported by [21,41], the concentration of the stock colloidal solution of the synthesized SNPs (1.7 μL/mg) was determined using the formulae presented by Liu et al. [42].

### 2.3. Characterization of SNPs

Both visually and spectroscopically, UV spectroscopy (Nano Drop ND2000, Thermo Scientific, Waltham, MA, USA) in the wavelength range of 300–800 nm with a resolution of 1 nm was used to observe the synthesis of the SNPs. The control sample was the autolyse without silver nitrate. To extract the biosynthesized SNPs, 13,000× *g* centrifugation for 1 h was utilized. For mass determination, the resulting nanoparticles were purified by washing with double-distilled water, organic solvents (ethanol), and drying at 40 °C. The size and shape of the SNPs were determined using TEM (JEM-1230, JEOL, Tokyo, Japan), which operated at a 120 kV acceleration voltage. After application, the SNPs were re-suspended in a carbon-coated copper grid with a 400 m mesh size, molecular grade sterile water, and allowed to dry at ambient temperature. Fourier transform infrared spectroscopy (FTIR; PerkinElmer) was used to depict surface composition variations in more detail [22,43]. The SNPs were pelletized using potassium bromide (KBr) at 1% (*w*/*w*) concentration. The SNPs were crushed for one minute at a pressure of ten tons, followed by FTIR analysis, to produce a clear pellet. Using dynamic light scattering (DLS), the average size and surface charge of the SNPs were determined [44]. Prior to analysis, the SNP sample (1 mg/mL) was diluted 100-fold with MiliQ water and ultrasonically treated to guarantee nanoparticle uniformity. The material was then examined using a Malvern DLS instrument (Nano-Zeta Sizer-HT, Malvern, UK).

### 2.4. Antibacterial Assays

#### 2.4.1. Isolation and Identification of Oral Pathogenic Bacterial Strains

Clinical pathogenic bacteria were isolated from several gingival sulcus specimens kindly taken from the oral cavity from dental clinics in Taif City, Saudi Arabia. The isolates were sub-cultured on mannitol salt agar and nutrient agar plates. Colonies with a clearly distinct morphology were picked and streaked over the same medium and cultured at 37 °C for 24–48 h. This procedure continued until single pure cultures were obtained. Colony and cell morphology, catalase production, oxidase activity, motility, and Gram reaction confirmed the isolates to the genus level. To characterize the 16S rDNA gene, a partial sequence amplified using genomic DNA as a template and the bacterial universal primers 27F (5′-GAGTTGATCACTGGCTCAG-3′) and 1492R (5′-TACGGCTACCTTGTTACGACTT-3′) were employed. To sequence the purified PCR products, Microgen Co, Seoul, Korea, used the 518F and 800R sequencing primers. Using the NCBI/BLAST tool, the obtained sequences were compared to the reference bacterial species in the gene-bank database. The nucleotide sequence data were deposited, and the accession numbers were obtained from the EMBL nucleotide sequence database.

#### 2.4.2. Susceptibility Testing

Susceptibility to 13 antibiotics was assessed using the standard Kirby–Bauer disc diffusion technique, as approved by the Clinical and Laboratory Standard Institute [45]. The SNPs from strain MS-37 were tested for biological activity against selected gingival pathogens using clinical and Laboratory Standards Institute-recommended fold broth microdilution procedure (CLSI) guidelines [45]. Bacteria were grown in Tryptic Soy Broth (TSB; Becton Dickinson, USA) for 24 h at 37 °C while being shaken at 125 rpm prior to the experiment. The assays were run in triplicate in 96-well plates. The range of the SNPs under investigation was 0.25 to 256 μg/mL, and the final bacterial concentration in each well of a plate was 5 ×10^5^ colony forming units (CFU) per ml. Both positive (inoculated medium) and negative (uninoculated media) controls were maintained. Multi-well plates with inoculation were incubated for 24 h at 37 °C. The lowest SNP concentration that prevented bacterial growth was identified visually as the minimum inhibitory concentration (MIC) values. By placing 100 μL of each test sample onto Tryptic Soy Agar (TSA; Becton Dickinson, USA) plates and incubating at 37 °C for 24 h, the minimal biocidal concentration (MBC) values of the SNPs were ascertained. The MBC was found to include the fewest SNPs that could stop >99.9% of bacterial cell development.

### 2.5. Antibiofilm Activity

The biofilm development experiment in microtiter plates was performed using the procedures described by [46]. TSB (100 μL) was loaded into 96-well flat-bottomed MTPs (Greiner Bio One, Fricken hausen, Mannheim, Germany) with/without supplements. Each well was supplemented with a diluted overnight bacterial culture (1:100 in TSB, 100 μL TSB). The positive control wells contained TSB injected with *S. aureus* ATCC29213, while the negative control wells contained just TSB. Plates were incubated at 37 °C for 18 h before being washed multiple times with PBS (pH 7.3). Sodium acetate (2%) was added as a fixative and decanted, and crystal violet (0.1 percent *w*/*v*) was used to stain the wells. Finally, the plates were washed under running tap water, dried, and read with a Sunrise absorbance reader at 570 nm (Tecan Austria GmbH, Salzburg, Austria). Antibiofilm activity was determined following the classification reported by Stepanovic et al [47].

### 2.6. Nitric Oxide Radical Scavenging Activity

The Griess reaction, which was based on [48], was used to assess the ability of the SNPs to scavenge nitric oxide radicals. The Griess reagent was prepared from equivalent amounts of 0.1% N-(1-naphthyl) ethylenediamine dihydrochloride and 1% sulphanilamide generated in 2.5% phosphoric acid. For this test, 500 μL of 10 mM sodium nitroprusside in phosphate-buffered saline (pH 7.4) and 1 mL of the SNPs at various concentrations (50–300 μg/mL) were incubated for 150 min at 25 °C. Then, 1.5 mL of freshly made Griess reagent was added to the product solution after the incubation period. The combination’s absorbance was then measured at 546 nm. The control and standard samples were made in the same way as the test samples by using buffer and Trolox instead of the SNPs. The ability of each extract to remove nitric oxide was evaluated [49]. The IC_50_ value is the concentration of the tested drug required to scavenge 50% of the nitric oxide radicals.

### 2.7. Inhibition of Protein Denaturation

The SNPs were examined for their influence on protein denaturation using the method published by Hmidani et al. [50] with the SNPs (1 mL) at various concentrations (50–300 μg/mL) and 1 mL 1% bovine albumin produced in phosphate-buffered saline (PBS, pH 6.4). The reaction mixtures were kept at 37 °C for 20 min and then heated to 70 °C for 5 min in a shaking water bath. After cooling, the turbidity of the reaction mixture was evaluated at 660 nm. The control and standard samples were produced similarly to the test samples, but they included buffer and diclofenac sodium (50–300 μg/mL) rather than the SNPs. According to Vijayakumar et al [51] findings, the percentage of inhibition of protein denaturation (% IPD) was calculated.

### 2.8. Cytotoxicity Assessment

Following treatment, to assess the substance’s cytotoxicity, the Alomar blue test was employed on tumor and non-tumor cells, as reported by Ahmed et al [52]. On 96-well plates, tumor and non-tumor cells were seeded at a density of 0.7 × 10^5^ cells/mL in 100 μL of complete media. After dissolving the SNPs in 0.5% DMSO, 50 to 300 g/mL was applied to the plates and incubated for 72 h. Following incubation, 20 μL l of Alomar blue stock solution (0.312 mg/mL) was given to each well. Using a microplate reader, the absorbance was measured at a wavelength of 570 nm (Molecular Devices; Sunnyvale, CA, USA). Using duplicate samples, at least three separate experiments were used to confirm each outcome. GraphPad Prism 7.0 (GraphPad Software Inc., La Jolla, CA, USA) was used to estimate the SNPs’ half-inhibitory concentrations (IC_50_).

### 2.9. Evaluation of SNPs on the Membrane of Guided Tissue Regeneration

Because biodegradable collagen membranes are routinely used for directed tissue regeneration, collagen was chosen as the substrate for nanosilver impregnation. This study used two GTR membranes (Perio Col^®^- GTR, Educare, Chennai, India): the GTR-C: plain GTR membrane as a negative control; the GTR-NS: GTR membrane impregnated with the SNPs as the test group. The GTR-NS membranes were prepared following the methods described previously [53,54]. Control membranes were frozen after 3 h in distilled water. After the treatment, the membranes turn light to dark yellow. Freeze-drying protects the material’s physical structure for storage or transport. Drying nanosilver [55] does not influence the drug release profile or antibacterial action.

For the qualitative assessment of microbial adhesion, membranes were cut into circular pieces 6 mm in diameter in a sterile setting. Membranes were put into Eppendorf tubes holding 2 mL of brain heart infusion broth (BHIB) media. All tubes were horizontally oriented. The tubes were injected with bacterial cultures of the tested oral pathogenic strains (3 × 10^4^ CFU/mL) and incubated at 37 °C. Bacterial adherence to the membranes was assessed after 1, 3, 5, and 7 days following the assessment criteria [56].

Penetrability studies were conducted in a device developed by [8] under a laminar flow hood. Each membrane was placed over an inner glass tube containing the growth material utilized in this investigation for each of the four strains. Silicone O-rings were used to seal the inner glass tubes, then inserted into the outer glass tube (Figure 1). The outer bottle was inoculated with a new culture of oral pathogenic bacterial strains (3 × 10^4^ CFU) in 3 mL of BHIB media, and the devices were incubated in the appropriate atmosphere. The bacterial counts in the inner tubes were determined at certain time intervals (1, 3, 5, and 7 days) using agar plates with the appropriate agar. The presence of colonies in the outer tube was verified on agar plates to avoid misrepresentative negative and positive findings. On BHI agar plates, colonies of various bacteria were counted.

### 2.10. Statistical Analysis

The data were interpreted using the Minitab statistical software (19.2020.1, Minitab Inc., Chicago, IL, USA). A *p*-value of < 0.05 denotes statistical significance.

## 3. Results and Discussion

Dentistry faces a significant challenge in oral health management because of the complexity of systems that prevent and control the spread of several microorganisms [57]. Since plaque allows bacteria to colonize teeth and is associated with several oral infectious diseases, plaque is a critical biological habitat [58]. The emergence of antibiotic-resistant bacteria, as well as the increasing frequency of hospital illness outbreaks have rekindled interest in non-pharmaceutical alternatives to synthetic therapies [59,60]. Since SNPs have excellent antibacterial resistance, they have found various applications [61]. The present work attempted to synthesize SNPs in an eco-friendly from an actinobacterium and investigated their biological activities such as antibacterial, antibiofilm, and anti-inflammatory. The actinobacterium isolate used in this study was identified molecularly using 16S rRNA. The MS-37 strain contains 97.62% *Streptomyces rochei*. The BLAST analysis and phylogenetic relationship to *Streptomyces rochei* strain MS-37 revealed a high similarity to the *Streptomyces rochei* SCSIO ZJ89 strain (MF 104551) (Figure 2).

The SNPs were identified and characterized using UV-Vis spectroscopy. In the reaction mixture, the results confirmed the presence of a peak with a maximum absorbance at 428 nm (Figure 3), which falls within the wavelength range recommended for SNPs and, thus, demonstrated their presence [22]. Additionally, various investigations revealed that SNPs were typically detected using UV-Vis spectroscopy, with peaks spanning 420 to 450 nm [62,63]. The SNPs were also spherical and polydispersed, with sizes ranging from 15–35 nm (mean size = 23.2 nm), as shown in Figure 4A,B, which was obtained using transmission electron microscopy. Nanomaterial size is significant because it affects its physical characteristics, cell penetration, and interactions with living cell molecules. Smaller silver nanoparticles have a higher surface area than larger particles when comparing the same amount of material, and their surface activity is higher as well [64]. The smaller the nanoparticles, the easier it is for them to pass through biological membranes and cause damage [7,20,22]. Contradicting the latter, [65,66] showed that gold nanoparticles with a diameter of 50 nm cross the cellular membrane more efficiently than nanoparticles with diameters of 30 nm and 14 nm, respectively, and [66] showed that gold nanostars with a total encumbrance of 75 nm enter cells better than nanoparticles with a diameter of 45 nm.

FTIR analysis was used to identify the biomolecules participating in the reduction of silver ions (Ag+) and the capping of the resulting SNPs. The FTIR spectra of the SNPs had six absorbance bands; 3400, 2925, 1640, 1385, 1013, and 690 cm^−1^ (Figure 5). The peak at 3400 cm^−1^ is attributed to the stretching vibrations of O-H bonds in alcohols and phenols [67]. The band at 2925 cm^−1^ (C–H stretch) belongs to the alkanes group, but the peak at 1640 cm^−1^ belongs to the N-H bend of primary amines [68]. The peak at 1385 cm^−1^ is attributed to symmetrical carboxyl group stretching [69]. The 1013 cm^−1^ band is related to the C–N stretching vibrations of aromatic and aliphatic amines [70]. FTIR data clearly demonstrated the presence of phenolic compounds and proteins that are likely engaged in the SNPs, as well as the potential that proteins play a significant role in the stabilization of the SNPs by capping, which inhibits agglomeration and helps to strengthen the stability of the SNPs [38].

In view of the fact that TEM images are captured using a dry sample and a high vacuum, additional DLS experiments were performed to determine the particle size in aqueous or physiological conditions. Therefore, DLS and Zeta were used to determine the particle size and potential stability of the SNPs. The SNPs had a particle size of 59.4 mm (Figure 6) and a Zeta potential of −26 mV (Figure 7), according to the obtained results. The SNPs had a particle size of 59.4 nanometers, which is somewhat larger than the particle size identified by TEM, likely owing to Brownian motion. Due to their encapsulation in an organic layer, the nanoparticles did not aggregate despite the fact that the SNPs agglomerated. Consequently, the size difference between the biosynthesized SNPs measured by TEM (23.2 nm) and DLS (59.4 nm) may be attributed to the fact that the two methods are based on fundamentally different physical principles. TEM analysis identifies the diameter of dried particles and the diameter of their metallic core, while DLS analysis measures the hydrodynamic radius of nanoparticles in solution, and the resultant nanoparticle size is always larger [71].

In most cases, bacteria that prevent the formation of periodontal pockets are the cause of periodontal infections [5]. Microbes thrive in periodontal pockets because they provide an ideal environment for them to survive and grow [72]. Dental cleanliness, pocket depth, the flow of gingival crevice fluid, gingivitis severity, type of interacting bacteria and viruses, host immune response, emerging pathogens, and antibiotic resistance all have an impact on the quantity and variety of microorganisms in the mouth [73]. As a result of their low cost and high effectiveness, antibiotics have traditionally been used to treat bacterial infection [55]. Several studies have found that widespread antibiotic use has resulted in the emergence of multidrug-resistant (MDR) bacterial strains. Clearly, antibiotic overuse has recently resulted in the emergence of MDR to nearly all antibiotics [21,74]. As a result, novel antimicrobial agents that are highly effective, non-invasive, non-toxic, and drug-resistant are required [75]. Surprisingly, nanoparticles are being evaluated as a possible alternative to antibiotics, and they appear to offer substantial promise in the fight against microbial MDR [18,76]. In light of this, gingival bacterial pathogens were isolated, and their antibiotic susceptibility was determined from the oral cavity of affected individuals. Additionally, the antibacterial efficacy of the SNPs produced was examined. The oral clinical isolates recognized the partial 16S rRNA gene sequence (continuous stretches of approximately 700–1145 bp). The resulting partial 16S rRNA gene sequences were deposited in the EMBL database. The bacteria isolated were identified as *Staphylococcus aureus* M0601, *Staphylococcus aureus* M0901, *Staphylococcus aureus* M1102, *Staphylococcus epidermidis* M0201, and *Staphylococcus hominis* M0401, using the accession codes shown in Table 1.

Various studies have shown that pathogenic bacteria acquired defense strategies that made them more difficult to treat, such as resistance genes or genetic alterations, resulting in extended infection with a greater mortality rate [4,75,77]. Nosocomial diseases have evolved from easily treated bacteria to highly resistant bacteria due to the widespread use of antimicrobial drugs. This shift presents a significant challenge for nosocomial infection control and prevention [21,78]. The term MDR bacteria refers to bacteria resistance to many antibiotics that they would ordinarily be sensitive to or to all antibiotic classes except for one or two [79]. Herein, the antibiotic susceptibility patterns of the selected bacteria were studied using the disc diffusion method, as detailed in Table 2. *S. hominis* M0401 and *S. epidermidis* M0201 showed the greatest antibiotic resistance patterns assessed. In addition, all isolates had a higher prevalence of MDR (n = 5−13) [21,74,78]. Interestingly, nanoparticles are now being evaluated as a possible alternative to antibiotics, and they appear to offer substantial promise in the fight against microbial MDR [18,76]. SNPs are a significant metallic nanoscale substance with significant antibacterial activity against various pathogens, including oral bacteria [20]. In this experiment, the biosynthesized SNPs showed an antibacterial impact against oral pathogenic bacteria, as revealed in Table 2. All bacterial isolates were highly sensitive to the SNPs, with MIC values ranging from 8 to 128 μg/mL. *S. epidermidis* M0201 was significantly less sensitive to the SNPs tested (MIC = 128 μg/mL and MBC = 256 μg/mL) than *S. aureus* M1102, *S. aureus* M0601, *S. aureus* M0901, and *S. hominis* M0401 (MIC values of 8, 16, 64, and 64 μg/mL, respectively; MBC values of 32, 32, 64, and 128). While SNPs’ antibacterial mechanisms of action have been widely studied and disputed, they are still not completely understood. SNPs have two well-established antibacterial mechanisms: direct and ion-mediated degradation [80]. When bacteria are exposed to SNPs, the nanoparticles bind to the cell wall’s surface [81]. SNPs have significant potential for efficiency enhancement by optimizing their physicochemical properties, also leading to a rise in the ability of bacteria’s macromolecules with functionalized sulfur and phosphorous to attach, causing cell death [82,83]. As a result, SNPs degrade the lipid bilayer’s integrity and the accessibility of the cytoplasmic membrane, as these are essential for the proper transport regulation via the cytoplasmic membrane [84,85]. Furthermore, SNPs’ antibacterial action produces a high amount of reactive oxygen and free radical species that prevent cell respiration and reproduction [86,87].

Furthermore, nanoparticles’ biocidal effects are aided by the silver ions they produce [88,89]. They might also affect potassium ion release and transit across microbial cell membranes. Ions, proteins, reducing sugars, and adenosine triphosphate are examples of cellular constituents (ATP), the cell’s energy reserve, leaking out of the cell due to the membrane’s increased permeability [90,91,92]. SNPs and/or silver ions can interact with biological components such as ribosomes and macromolecules including proteins, lipids, and DNA in microbial cells, killing the organisms. They prevent the function of proteins, the translation of ribosomes, and DNA replication [88,93].

Antibiotic resistance is often prevalent at a high to moderate level, resulting from their biofilm-forming ability [94]. As a result, inquiries into the pathogenesis of these diseases have concentrated on the method by which these microbes adhere to the collected specimens. Their high incidence of antibiotic resistance may result from their ability to form biofilms. All isolated *Staphylococcus* spp. could form a biofilm on polystyrene surfaces, although in various patterns (Table 3). *S. aureus* strains M1102, *S. hominis* M0401, and *S. epidermidis* M0201 are strong biofilm producers (+++), but *S. aureus* M0901 exhibits a moderate ability to form a biofilm (++), while *S. aureus* M0601 is a weak biofilm producer (+). Regarding antibiofilm activity, SNP treatment had a significant effect on the majority of strains, lowering their ability to form biofilms from strong (+++) to non-producing (-). On the other hand, the SNPs had no influence on *S. epidermidis* M0201 adherence to biofilms (Table 3). It is reported that strain-dependent inhibition of staphylococcal biofilm formation was detected when the culture media was supplemented with the SNPs. Additionally, these biosynthesized SNPs had significant antibiofilm action against pathogenic bacteria related to gingival disease.

Inflammation is a physiological response to potentially harmful stimuli such as irritants, damaged cells, or infection [11]. Systemic or localized inflammation can be acute or chronic [12]. To protect cells and tissues during the acute inflammatory process, different immune cells or neutrophil respiratory bursts produce a wide range of mediators, including prostaglandins, cytokines, and other ROSs, such as nitric oxide (NO) [12]. The antioxidant activity of standard Trolox and the SNPs was evaluated using a nitric oxide radical scavenging test, with IC_50_ values of 110.7 ± 6.15 and 80.07 ± 4.2 μg/mL, respectively (Table 4). The SNPs may be essential in minimizing the adverse effects of excessive NO creation in the human body since they can reduce NO production. Additionally, the scavenging activity might obstruct the series of negative effects brought on by excessive NO creation [95]. Increased vascular permeability, protein denaturation, and membrane alteration are just a few of the many processes that contribute to inflammation, which frequently causes modification. Denaturation occurs when proteins lose their tertiary and secondary structures due to stress or heat. The IC_50_ values for diclofenac sodium and the SNPs were determined to be 215.5 ± 4.90 μg/mL and 189.44 ± 5.52 μg/mL, respectively (Table 4). reported similar findings, which might be explained by the combined impact of bioactive agents adsorbed on the surface of SNPs, which increases their dispersibility and bioavailability [96].

Cytotoxic activity is a critical feature to consider when determining a substance’s safety for usage in medical uses. Therefore, to assess the tumor cytotoxicity of the SNPs after 72 h of treatment, a cytotoxicity test using Alomar Blue was performed. The investigation used normal mammalian cells, peripheral blood mononuclear cells (PBMC), and an oral adenosquamous carcinoma cell line (CAL27). The SNPs had an IC_50_ of 81.16 μg/mL in PBMCs and 34.03 μg/mL in CAL27 (Table 4). The toxicity of the SNPs in PMBCs may be attributed to the release of free silver ions, the total silver ion concentration, or the interaction of cellular components with the nanoparticles. Furthermore, SNPs have demonstrated a range of cytotoxic effects in several cell types, indicating that they impair cell viability by interfering with mitochondrial structure and metabolism [62].

The adhesion and penetration of the chosen strains into their corresponding membrane groups were tested in this investigation during four-time intervals. As shown in Table 5, the mean adherence score increased significantly at the end of Days 3, 5, and 7 for the two groups compared to Day 1 (*p =* 0.001). The SNPs significantly reduced *S. epidermidis* M0201 adhesion to the membrane in GTR-NS, reaching a maximum of 1.8 ± 0.43 after day 7, compared to 3.3 ± 0.21 for GTR-C, a difference of *p* ꞊ 0.001. The mean bacterial adherence scores were significantly higher in the GTR-C group than in the GTR-NS group throughout various incubation times and bacterial strains. This difference was statistically significant with respect to the adherence scores (*p* ꞊ 0.001). This experiment employed collagen membranes as the substrate for silver nanoparticle deposition. Bacterial adhesion is observed to decrease when the hydrophobicity of biomaterials rises [19,23]. Due to its greater hydrophilicity, collagen is more susceptible to bacterial colonization by *S. mutans*, *A. actinomycete mcomitans*, *F. nucleatum*, and *P. gingivalis* than other GTR membranes [8]. Collagen is a viscoelastic substance with a high tensile strength, but limited elasticity [96] or loading with other nanoparticles may affect their fundamental physical characteristics [97,98]. The mean CFUs cultivated from the inner tube were used to determine bacterial penetration across the GTR membranes (Table 6). The number of CFU/mL cultured from the inner tube on Days 3, 5, and 7 was contrasted with the number of CFU/mL cultured from the inner tube on one day. In terms of penetration for all examined bacteria, the mean CFUs grown from the inner tube were higher in the GTR-C group than in the GTR-NS group at all incubation intervals. The number of CFU/mL cultivated from the inner tube significantly decreased (P 0.001) on Days 3, 5, and 7 in the GTR-NS group, *S. aureus* M1102, M0601, and M0901. Additionally, no growth was seen in the inner tube culture of *S. hominis* M0401 or *S. epidermidis* M0201 (Table 6). A GTR membrane that is medically controllable must be both rigid and elastic [97]. Suppose SNPs-coated GTR membranes are approved for intraoral clinical use. In that case, it will be intriguing to observe how the change in mechanical characteristics impacts clinical manageability, space formation, and, eventually, the possibility of periodontal regeneration [7].

## 4. Conclusions

The findings of this work provide a new path for the application of marine-derived SNPs, which demonstrated outstanding antibacterial activity against clinically isolated oral biofilms from gingivitis specimens. Studies indicate that SNPs have anti-inflammatory effects by preventing protein denaturation and using nitric oxide to scavenge free radicals. While the present study’s in vitro results on the CAL27 cell line are encouraging, the efficacy of SNPs with known effects on viability should be investigated in vivo. This information will be critical for pre-clinical studies examining the therapeutic potential of SNPs in gingivitis therapies.

## Figures and Tables

**Figure 1 molecules-27-07296-f001:**
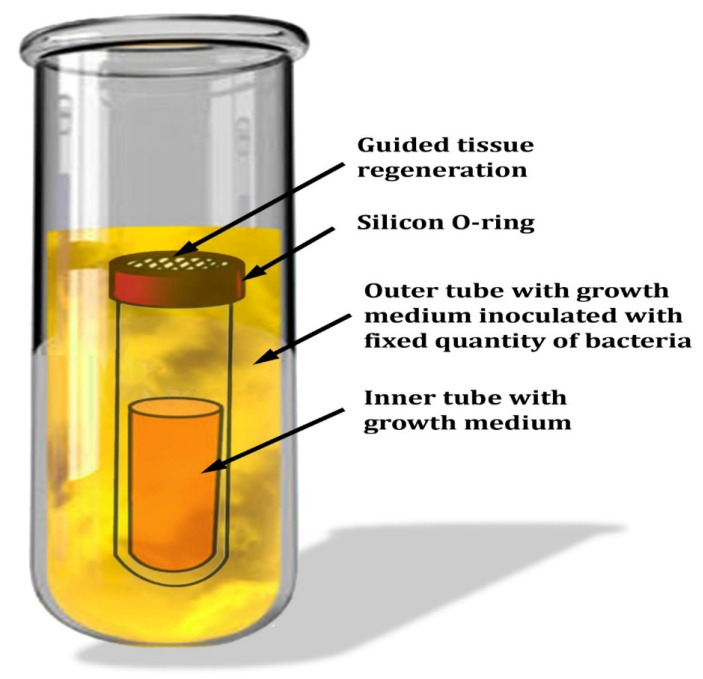
Schematic representation of the in vitro experimental setup for the permeability test of bacteria.

**Figure 2 molecules-27-07296-f002:**
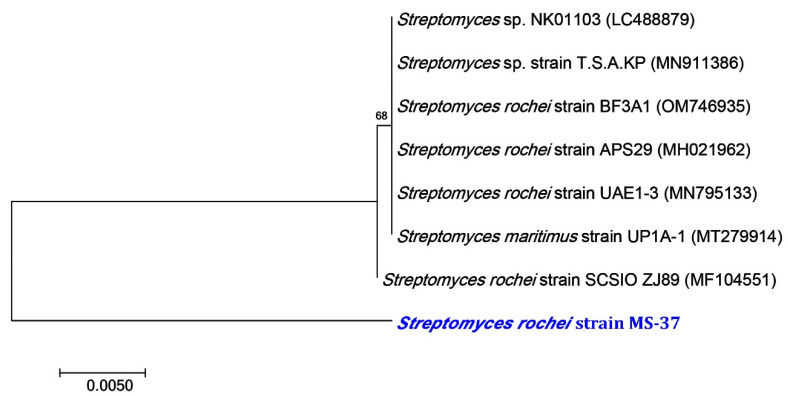
A neighbor-joining tree of *Streptomyces rochei* MS-37 with its closely related taxa. The bootstrap consensus tree inferred from 1000 replicates represents the evolutionary history of the taxa analyzed. The scale bar indicates 0.005 substitutions per nucleotide position.

**Figure 3 molecules-27-07296-f003:**
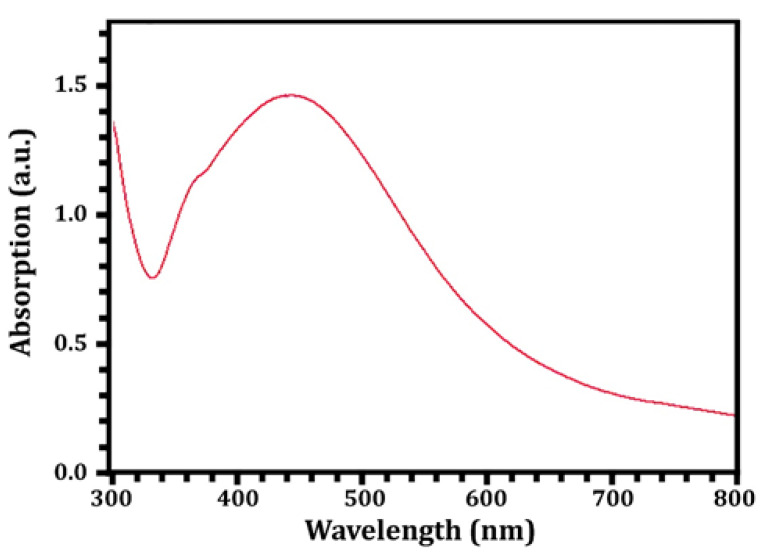
UV-Vis spectra of the *Streptomyces rochei* MS-37-derived SNPs.

**Figure 4 molecules-27-07296-f004:**
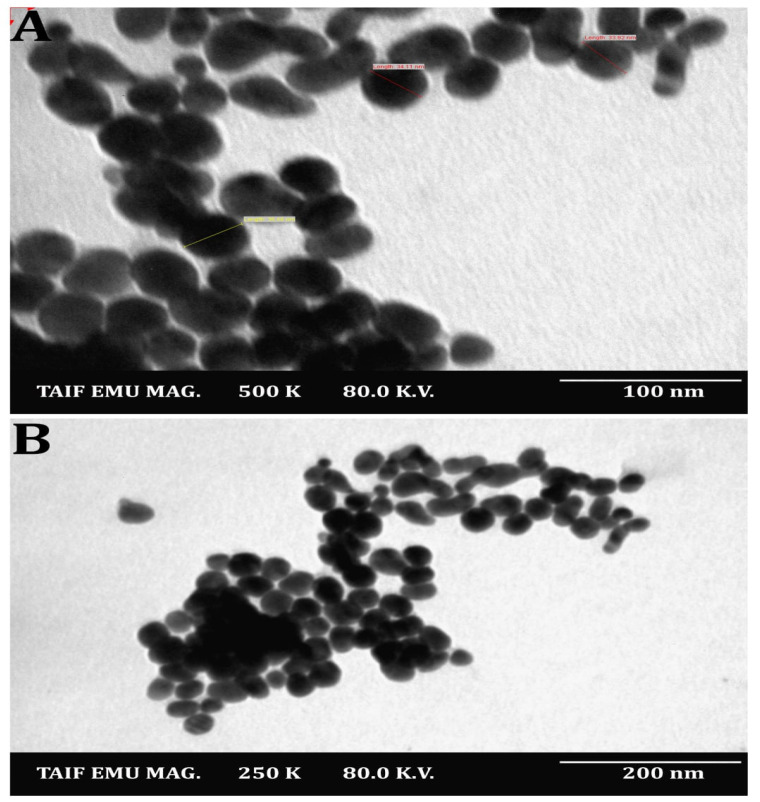
TEM images of the *Streptomyces rochei* MS 37-derived SNPs. Magnification power at 100 nm (**A**) and at 200 nm (**B**).

**Figure 5 molecules-27-07296-f005:**
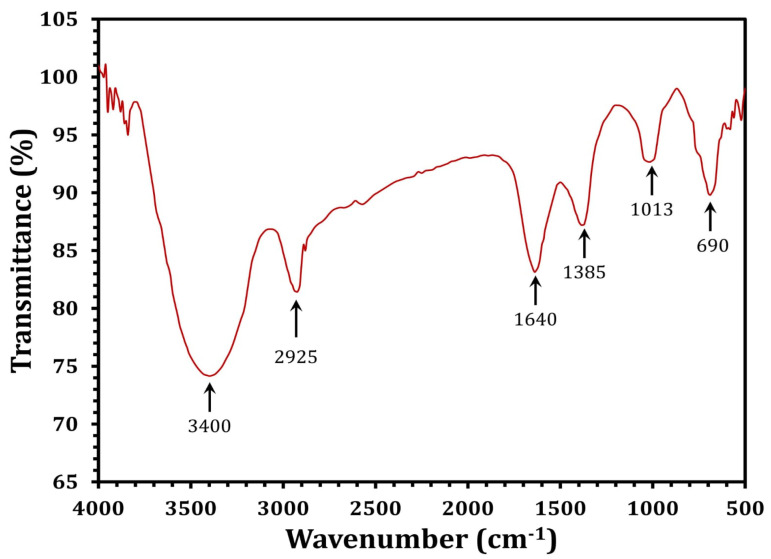
FTIR spectra of the *Streptomyces rochei* MS-37-derived SNPs.

**Figure 6 molecules-27-07296-f006:**
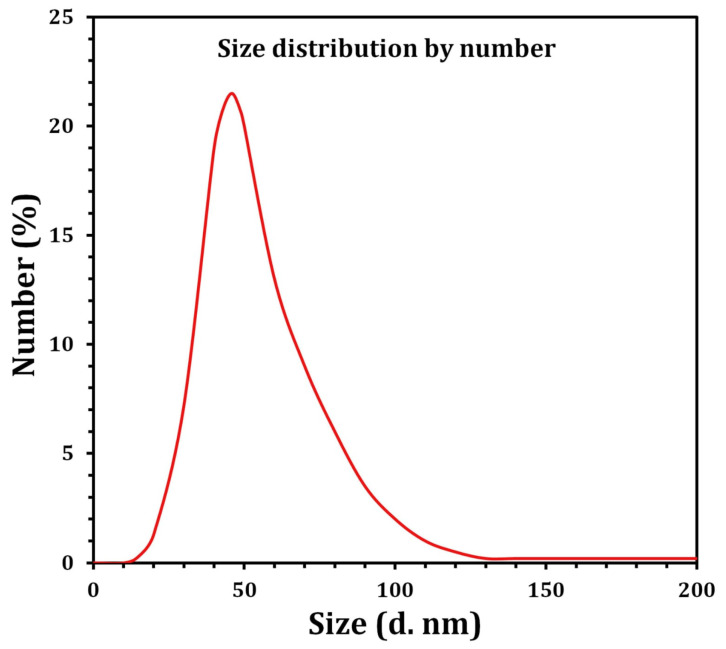
Size distribution by the number graph of the SNPs as revealed by DLS.

**Figure 7 molecules-27-07296-f007:**
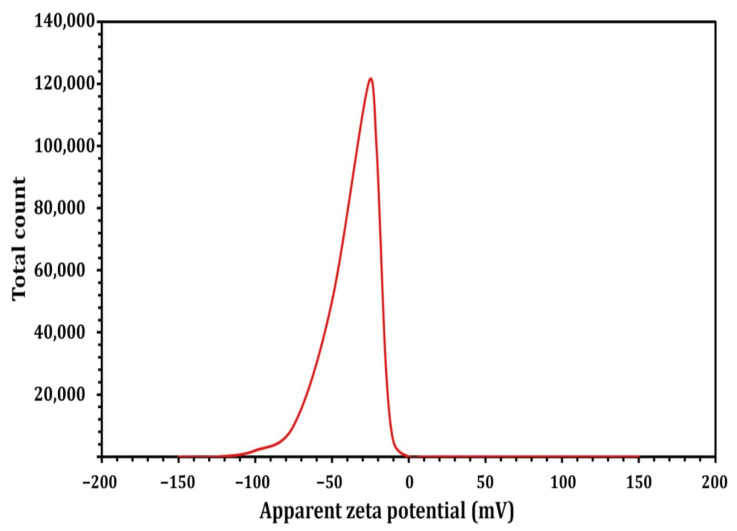
Zeta potential result of the *Streptomyces rochei* MS- 37 SNPs.

**Table 1 molecules-27-07296-t001:** Molecular identification of oral cavity bacterial species based on GeneBank BLAST comparisons.

Strain No.	Nomenclature	Accession No.	Closest Relative Yeast	Sequence Identity (%)
M0601	*Staphylococcus aureus*	LN899797	*Staphylococcus aureus* strain WO53 (LC107797)	100.0
M0901	*Staphylococcus aureus*	LN899798	*Staphylococcus aureus* strain GR41 (LC107809)	99.57
M1102	*Staphylococcus aureus*	LN899800	*Staphylococcus aureus* isolate H3 (LN899816)	99.90
M0201	*Staphylococcus epidermidis*	LN899795	*Staphylococcus epidermidis* strain TMPC 9023C (OM265429)	100.0
M0401	*Staphylococcus hominis*	LN899796	*Staphylococcus hominis* strain PL562 (MK015863)	99.68

**Table 2 molecules-27-07296-t002:** Antimicrobial activity of the SNPs against periodontal pathogenic strains.

Periodontal Pathogens	Antibiotic Resistance Pattern	^#^ SNPs (μg/mL)	
		MIC	MBC
*S. aureus* M1102	VA, E, PB, OX, CAZ	8	32
*S. aureus* M0601	TE, VA, E, PB, AMC, OX, FOX	16	32
*S. aureus* M0901	VA, GN, NOR, E, PB, AMC, OX, FOX, CAZ	64	64
*S. hominis* M0401	TE, VA, GN, CIP, NOR, E, C, PB, AMC, OX, CAZ	64	128
*S. epidermidis* M0201	TE, VA, GN, AK, CIP, NOR, E, C, PB, AMC, OX, FOX, CAZ	128	256

**TE**, Tetracyclines; **VA**, Vancomycin; **GN**, Gentamicin; **AK**, Amikacin; **CIP**, Ciprofloxacin; **NOR**, Norfloxacin; **E**, Erythromycin; **C**, Chloramphenicol; **PB**, Polymyxin B; **AMC**, Amoxicillin-clavulanic acid; **OX**, Oxacillin; **FOX**, Cefoxitin; **CAZ**, Ceftazidime. ^#^**SNPs**, silver nanoparticles; **MIC**, minimum inhibitory concentration; **MBC**, minimum bactericidal concentration.

**Table 3 molecules-27-07296-t003:** Quantitative detection of biofilm formation by Staphylococcus sp. using the MtP assay.

Bacterial Strain	Biofilm Formation before Treatment		Biofilm Formation after Treatment with SNPs	
	Producing Category	*A* _630nm_	Producing Category	*A* _630nm_
*S. aureus* M1102	+++	0.272 ± 0.00	-	0.10 ± 0.002
*S. aureus* M0601	+	0.110 ± 0.01	-	0.08 ± 0.003
*S. aureus* M0901	++	0.210 ± 0.01	-	0.07 ± 0.004
*S. hominis* M0401	+++	0.255 ± 0.00	-	0.11 ± 0.001
*S. epidermidis* M0201	+++	0.56 ± 0.02	+++	0.45 ± 0.002

-, not a producer; +, weak; ++, moderate; +++, strong biofilm producer.

**Table 4 molecules-27-07296-t004:** Effect of the SNPs on nitric oxide radical scavenging activity, protein denaturation inhibition, and cytotoxicity (μg/mL).

Tested Materials	IC_50_ (μg/mL)			
	Nitric Oxide Radical Scavenging Activity	Inhibition of Protein Denaturation	* Cytotoxicity	
			CAL27	PMBC
SNPs	110.7 ± 6.15	89.44 ± 5.52	34.03	81.16
Standard	80.07 ± 4.2 ^#^	215.5 ± 4.90 ^##^	-	-

The data are provided as IC_50_ values in μg/mL with a 95% confidence interval derived from at least three independent tests conducted in duplicate. * Cytotoxicity was determined 72 h after treatment using the Alamar Blue test. SNPs, silver nanoparticles; CAL27, oral adenosquamous carcinoma cell lines; PMBC, human peripheral blood mononuclear cells. **^#^** Trolox; **^##^** diclofenac sodium.

**Table 5 molecules-27-07296-t005:** Qualitative assessment of bacterial adherence.

Group	Day	Adherence Score				
		*S. aureus* M1102	*S. aureus* M0601	*S. aureus* M0901	*S. hominis* M0401	*S. epidermidis* M0201
GTR-C	1	1.1 ± 0.3	1.0 ± 0.0	1.1 ± 0.34	1.3 ± 0.6	1.4 ± 0.8
	3	1.99 ± 0.33 *	1.1 ± 0.3	1.2 ± 0.24	1.5 ± 0.5	2.05 ± 0.2 *
	5	2.04 ± 0.24 *	2.2 ± 0.45 *	2.14 ± 0.4 *	3.0 ± 0.24 *	3.1 ± 0.3 *
	7	2.9 ± 0.3 *	3.2 ± 0.1 *	2.95 ± 0.23 *	3.1 ± 0.1 *	3.3 ± 0.21 *
GTR-NS	1	0.08 ± 0.1	0.05 ± 0.01	0.08 ± 0.04	0.9 ± 0.1	1.0 ± 0.02
	3	1.0 ± 0.2 *	1.1 ± 0.4 *	1.3 ± 0.5 *	1.3 ± 0.5 *	1.4 ± 0.3 *
	5	1.5 ± 0.3 *	1.4 ± 0.6 *	1.6 ± 0.4 *	1.4 ± 0.2 *	1.6 ± 0.4 *
	7	1.8 ± 0.8 *	1.5 ± 0.72 *	1.7 ± 0.48 *	1.5 ± 0.5 *	1.8 ± 0.43 *

* Highly significant as compared to the levels at Day 1 using the Tukey HSD/multiple comparisons. Values are the mean of the triplicate ± SD. GTR, guided tissue regeneration; C, control; NS, nanosilver. *p*-value < 0.05 is considered significant.

**Table 6 molecules-27-07296-t006:** Bacterial penetration of control (GTR-C) and silver-nanoparticle (GTR-NS)-impregnated collagen guided tissue regeneration (GTR) membranes.

Group	Day	CFU/ml				
		*S. aureus* M1102	*S. aureus* M0601	*S. aureus* M0901	*S. hominis* M0401	*S. epidermidis* M0201
GTR-C	1	285.2 ± 3.0	125.0 ± 0.0	95.1 ± 0.0	121.17 ± 3.8	160.4 ± 0.8
	3	255.5 ± 2.9 *	155.58 ± 0.3	198.6 ± 1.2	199.8 ± 1.6	210.2 ± 0.2 *
	5	240.1 ± 0.24 *	278.2 ± 8.45 *	198.8 ± 0.4 *	235.17 ± 6.24 *	270.0 ± 5.3 *
	7	179.58 ± 7.3 *	269.4 ± 4.51 *	300.0 ± 3.3 *	285.2 ± 3.1 *	330 ± 4.81 *
GTR-NS	1	146.78 ± 8.1	99.0 ± 0.1	82.7 ± 1.4	90.0 ± 8.1	100.0 ± 7. 2
	3	126.54 ± 9.2 *	84.9 ± 3.4 *	71.2 ± 6.5 *	71.2 ± 4.5 *	94.2 ± 5.3 *
	5	115.0 ± 0.3 *	70.0 ± 6.6 *	51.4 ± 4.4 *	54.0 ± 2.2 *	46.0 ± 4.4 *
	7	74.47 ± 10.8 *	40.8 ± 7.72 *	32.17 ± 6.8 *	0.0 ± 0.0 *	0.0 ± 0.0

* Values are the mean of the triplicate ± SD. *p*-value < 0.05 is considered significant.

## Data Availability

This article has all the data that were created or evaluated during this investigation.

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
