# Peer review of "Streptomyces rochei MS-37 as a Novel Marine Actinobacterium for Green Biosynthesis of Silver Nanoparticles and Their Biomedical Applications"

_molecules, 2022, doi:10.3390/molecules27217296_

Round 1

Reviewer 1 Report (Previous Reviewer 1)

I read the revision, in my opinion the paper can now be published

Reviewer 2 Report (Previous Reviewer 2)

Authors have addressed all my comments.

This manuscript is a resubmission of an earlier submission. The following is a list of the peer review reports and author responses from that submission.

Round 1

Reviewer 1 Report

The manuscript is interesting, especially on the microbiological point of view, that has been thoroughly investigated with a variety of up-to-date techniques and that demonstrated of membranes impregnated with silver nanoparticles for guided tissue regeneration in dental diseases. However, the introduction, general discussion and nanotechnological part are weak and need revision. In particular, these points must be stressed and revised

1)      Abstract: the acronym PBMC is not defined

2)      Introduction, lines 82-95: among the SNP syntesized with green approaches and with great potential for dental treatment, this paper showing the efficency of SNP + fluoride ion should be cited: doi:10.3390/molecules25153494. Moreover, regarding the use of SNP as antimicrobials, beside extended self-citation of their research papers (refs 15-21), the authors must cite general reviews on the topic, among which Angew. Chem. Int. Ed., 2013, 52, 1636– 1653; Int. J. Antimicrob. Agents, 2009, 34, 103– 110; Eur. J. Inorg. Chem. 2018, 4846–4855

3)      Introduction, lines 101-104: the authors state that Streptomyces rochei MS-37has not yet been used yet for the green synthesis of SNP for antimicrobial applications. However, a quick search on Web od Science, allowed me to find this paper: Mabrouk, M; Elkhooly, TA; Amer, SK, Actinomycete strain type determines the monodispersity and antibacterial properties of biogenically synthesized silver nanoparticles”, 2021, JOURNAL OF GENETIC ENGINEERING AND BIOTECHNOLOGY 19 (1), in which the synthesis of SNP with Streptomyces rochei is reported. This paper must be cited. Moreover, what are the advantges of the synthesis with Streptomyces rochei with respect to the hundreds of green synthesis of silver nanoparticles reported in the literature? Authors should justify their choice for this particular synthetic approach and bacterial strain with respect to the many other experiments reported by literature

4)      Experimental, line 130 “ultraviolet-130 visible (UV) spectroscopy”. As the examined range is 300-800 nm, it is better to describe the experiment as UV-Vis absorption spectroscopy.

5)      Results and discussion, lines 273-285. All the discussion on SNP along this lines should be revised. Small silver nanoparticles (d < 40 nm) have a sharp LSPR absorption peak centered at 400-420 nm. For a series of absorption spectra examples see anc cite D.D. Evanoff Jr, G. Chumanov, ChemPhysChem 2005, 6, 1221-1231. The very large absorption band displayed in figure 4A is indicative of non spherical shapes or aggregated nanoparticles. As a matter of fact, this is exactly what is seen in figure 4B (TEM image) in which a lot of elongated, “bean-like” particles are clearly visible. Indeed, the statement “SNPs were also spherical”, line 276, is not corresponding to the spectral and TEM data. Moreover, Figure 4B does not contain a scale bar, so the real SNP dimensions cannot be evaluated by the readed (please add the dimensional bar!). Finally, the intuitive statement “The smaller the nanoparticles, the easier it is for them to pass through 282 biological membranes and cause damage” at lines 282-283 has been contradicted since a long time. Please mention and discuss the classical papers Nano Letters 2006, 6, 662-668 and Nano Letters 2007, 7, 1542-1550, that have demonstrated that spherical gold nanoparticles of diameter 50 nm crosses more efficiently the cellular membrane than nanoparticles with diameter 30 nm and 14 nm, and also a more recent example, on J. Colloid Interface Sci. 2017, 505, 1055-1064, that showed that gold nanostars with total encumberance 75 nm enter better in cells than the same nanoparticles of 45 nm

6)      The characterization of SNP is insufficient. Calculate and discuss the yield (% of Ag as SNP with respect to total starting Ag+). Describe and comment the reproducibility of the synthesis: how many times has it been carried out? Does it always yield the same dimensional and shape distribution of SNP? Add to the supplementary materials more TEM images (at least 2 for each different synthesis), taken from ate least 3 different preparations, so the reader can actually evaluate the reproducibility of the method. What is the stability of these SNP? What does it happen if they are suspended in classical highly saline media at physiological pH (7.4), like PBS? Do they remain as colloidal suspension or rapidly/slowly aggregate? What the SNP are coated of? If in the availability of the authors, a TGA (thermogravimetric analysis) should be carried out to understand the quantity of coating around the SNP

Reviewer 2 Report

1. Nanoparticle biosynthesis section is not complete. Some important information is missing, like grade of chemicals, reason of incubation, separation of nanoparticles, heating, etc.

2. It is difficult to differentiate between sheets with and without impregnation

3. UV of all samples should be provided

4. FTIR is important in this study, which is missing.

5. Zeta potential is missing.

6. Overall this article is incomplete and did not contribution sufficient information to the existing knowledge.